# A Review of System-in-Package Technologies: Application and Reliability of Advanced Packaging

**DOI:** 10.3390/mi14061149

**Published:** 2023-05-29

**Authors:** Haoyu Wang, Jianshe Ma, Yide Yang, Mali Gong, Qinheng Wang

**Affiliations:** 1Tsinghua Shenzhen International Graduate School, Tsinghua University, Shenzhen 518055, China; haoyu-wa21@mails.tsinghua.edu.cn (H.W.); yyd22@mails.tsinghua.edu.cn (Y.Y.); 2Department of Precision Instrument, Tsinghua University, Beijing 100084, China; gongml@mail.tsinghua.edu.cn; 3Shanghai Kulan Electronic Technology Co., Shanghai 200040, China

**Keywords:** integrated circuits, system-in-package, package reliability, advanced packaging, optimization

## Abstract

The system-in-package (SiP) has gained much interest in the current rapid development of integrated circuits (ICs) due to its advantages of integration, shrinking, and high density. This review examined the SiP as its focus, provides a list of the most-recent SiP innovations based on market needs, and discusses how the SiP is used in various fields. Reliability issues must be resolved if the SiP is to operate normally. Three factors—thermal management, mechanical stress and electrical properties—can be paired with specific examples in order to detect and improve package reliability. This review provides a thorough overview of SiP technology, serves as a guide and foundation for the SiP in package reliability design, and addresses the challenges and potential for further development of this kind of package.

## 1. Introduction

Moore’s Law, known as the “gold standard” of the chip industry, appears to have reached a “bottleneck” stage as a result of the ongoing advancements in integrated circuit technology [1,2,3,4]. Going Beyond Moore’s Law has been suggested in this context [5,6,7], and the industry has given system-in-package (SiP) technology much attention. SiP technology combines numerous active devices that are based on bare chips with various passive devices that are all combined into a single package. A system-level device capable of performing specific operations is ultimately created through the processing procedure [8]. It is clear from the definition that the SiP is situated at the packaging’s integration level. As indicated in Figure 1, there are three stages of electronic integration technologies now available: chip-level, package-level, and board-level [9]. The first level of integration is known as the system-on-chip (SoC), which is a deep integration of the system’s components onto a single chip. The second level of integration is known as the SiP, which is a side-by-side or stacking of many chips [10]. To accomplish particular goals, the packaging is performed in 2D, 2.5D, or 3D. The board level, also referred to as the printed circuit board (PCB), is the third level.

Table 1 [11,12] displays a comparison of the traits of the three levels. The SiP design is based on the SoC [13], and the PCB is also required to execute the functionality when the SiP design is finished. It should be noted that the three levels are complementary to one another. As a result, it is impossible to fairly compare the three’s performance in one particular area.

In fact, on the branch of packaging, besides the SiP, there is also the Chiplet [14]. The Chiplet, which is a small chip/core, is made by separating the components originally integrated in the same system monolithic wafer into multiple Chiplets with specific functions and then interconnecting them through advanced packaging technology to finally integrate the package into a system chipset. Chiplets are closely related to many advanced packaging technologies, so some people even consider Chiplets as an extension and expansion of the SiP.

In heterogeneous integration, SiP technology is particularly important to achieve the goal of “Beyond Moore’s Law”. The SiP has value in almost every market segment. According to the Yole Group [15], SiP’s share of hours exceeded USD 14B in 2020 and is expected to increase to at least USD 19B by 2026.

As shown in Figure 2, the SiP has evolved through three stages, namely original device packaging, SiP conventional packaging, and SiP advanced packaging. Original device packaging has DIP, LCC, SOT, and other inline or surface mount types of devices; SiP conventional packaging is the primary SiP packaging represented by flip chip sand wire bonding; in the new century, there are various more advanced packaging forms. It can be seen that the SiP is developing in the direction of systemization, functionalization, and diversification.

The SiP provides quick development cycles, high levels of flexibility, and low costs as compared to other packaging types, especially when it comes to optimizing the benefits of the system. The current increasing demand for high-performance packages has led to a preference for SiP technology [10], where a “combined” system may contain the functionality of many chips. The SiP is advancing in terms of density, power, and miniaturization along with the advancement of integration and packaging technologies [8,13]. SiPs include a variety of advanced packaging technologies such as the fan-out wafer-level package (FOWLP), chip-on-wafer-on-substrate (CoWoS), and others [14,16]. This review looked into some of the more common packaging techniques being used and identified several industrial domains where SiP technology is being used. The development of SiP technology has been accompanied by many reliability issues. To improve the overall performance of the SiP, thermal management, mechanical stress, and electrical properties need to be addressed [17]. This review looked into SiP design optimization techniques, as well as reliability fault detection schemes.

The goal of this review was to thoroughly examine the creation, cutting-edge technologies, applications, reliability analysis, and optimization of SiPs. It is broken down into a total of six sections: Section 2 introduces the SiP’s advanced packaging technology; Section 3 describes how it is used in various industrial fields; Section 4 provides the analysis and detection techniques for the SiP reliability issues; Section 5 discusses the idea of improving package reliability by optimizing the SiP design; Section 6 is a summary and outlook of the whole review.

## 2. Advanced Packaging Technologies in SiP

### 2.1. Interconnect

SiPs involve many interconnection cases, the main paths being wire bonding, solder joints and bumps, etc. Wire bonding is a method of connecting metal leads to the pads, which is equivalent to connecting the internal and external chips. From a structural point of view, the metal leads act as a bridge between the pads of the chip and the carrier pads. The former is the first bonding, and the latter is the second bonding. Solder joints are the conventional connection method and are more frequently used at the PCB level. Solder joints are usually used to connect the PCB or substrate with other electronic components. Bumps can be understood as reduced solder joints, often called a solder ball, which can be used to connect tiny devices.

The basic components of a SiP’s design schematic are a combined chip, a silicon intermediary layer [18], a substrate, and a PCB, as depicted in Figure 3. The chips are integrated in a 2.5D or 3D shape as a stack. The two versions differ in that wiring and punching holes are performed directly on the chip in the latter, while wiring and punching holes are performed on an intermediary layer, which is inserted between the chip and the substrate in the former, such as through silicon vias (TSVs).

Wire bonding and flip chips are the two more popular varieties in the standard SiP procedure. As seen in Figure 4, the die/chip is wire-bonded to the substrate by these two methods. Wire bonding is utilized to attach the die to the substrate’s top, and the flip chip is used to complete the interconnection by flipping down a conductive underfill, which has been placed right on the surface of the die to link it to the substrate. The flip chip is somewhat of an advancement over wire bonding, which primarily manifests in the following area: (1) Signal integrity (SI): The advantages of flip chips are more visible because the signal line path is significantly shorter than that of the wire bond, especially in high-speed signal scenarios with strict requirements for signal integrity [19]. (2) Power integrity (PI): The flip chip is smaller; the overall impedance from the power supply to the chip will be lower, reducing the voltage drop and power supply noise. (3) Flexibility: Flip chips have clear advantages in high-density connectivity situations, as they allow for connections throughout the entire die’s surface. (4) Miniaturization: Without additional room for connections, the high-density feature of the flip chip directly benefits system area reduction.

Sophistication and diversification have also been present in the development of SiP technology. There are three further common forms of SiPs, namely flip chip, fan-out, and embedded die, according to the Yole Group’s [15] analysis of system-level packaging technology and market trends. The fan-out and embedded die have clear competitive benefits over the flip chip packaging method in terms of structural volume, quantity, and the redistribution layer (RDL). They also perform well in terms of signal transmission and thermal management.

People have increased their demands for the manufacturing efficiency of the IC sector as a result of cost increases and technological advancements [20]. In order to increase the production efficiency of the entire sector, major corporations are gradually shifting toward a vertical division of the labor model in which design, manufacturing, packaging, and testing are separated. More importantly, the worldwide supply chain for SiP technology is extremely intricate and sophisticated, encompassing integrated design and manufacture (IDM), outsourced semiconductor assembly and testing (OSAT), and foundries. As a result, an increasing number of businesses are investing in research into integrated packaging, which keeps the SiP consistently inventive. Most of the OSAT companies, including ASE, Amkor, and JCET, which have achieved the top spot in packaging and testing, as well as the world’s top IDM companies, Samsung Electronics, Intel, and Infineon, work to improve product competitiveness through SiP technology, particularly in the fields of the fan-out and embedded die. Universities and research organizations have also made contributions to the theoretical aspects of cutting-edge SiP technologies.

It has been discovered through extensive literature and website study that SiP advanced technologies are diverse.

### 2.2. Fan-Out Wafer-Level Package

The wafer is often chopped and sliced before being packaged according to customary packaging methods. Wafer-level packaging (WLP) involves placing an entire package on a wafer and then slicing it once it is finished. With this method, the package size can be close to the same as the bare chip size, which is consistent with the idea of electronic items being made smaller and lighter. There are two types of WLP: one is the fan-out type, which may be rewired outside the area where the chip is located to gain more pins, while the other one is the fan-in type, where the package’s two-dimensional size is roughly equal to the chip’s own size. The fan-out type becomes a better option as IC output pin counts rise.

The overall system density for FOWLP containers is very high. In order to modify the wiring technique corresponding to the original chip pins, which can be applied to more varieties of package forms, it is typically used in conjunction with the RDL [21,22]. Figure 5 depicts an FOWLP package based on RDL technology [23]. It consists of eight stages, including deposition, RDL fabrication, assembly, etching, and cleaning. The structure is designed to be extremely flexible, both as an overall SiP connected to the board level via solder joints and cut into individual chips for use. Theoretically, the RDL metal rewiring and bumping method can match the fan-out type and achieve the ideal layout. FOWLP thus addresses the issue of inadequate implant balls caused by too many chip pins in conventional wafer-level packages.

### 2.3. Embedded Wafer-Level Ball Grid Array

The embedded wafer-level ball grid array (eWLB) is derived from the FOWLP [24]. Infineon created the first version of the eWLB by laying the qualified wafer face down on the larger wafer. The two are placed together into an epoxy mold before being finished with packing, wiring, dicing, etc. Many companies or organizations have improved and optimized their antenna packages after learning from Infineon’s design architecture [25,26]. In order to use the eWLB, which is a fifth-generation packaging technique, the chip must first be rearranged onto an artificial wafer, followed by fanning out to the chip’s quad-axis, ball planting, and finally, packaging. This type of packaging not only boosts route density, but also reduces the package size, freeing up much board space. More crucially, the shorter interconnects and lower dielectric constant materials also result in better electrical performance. The eWLB package type offered by JCET has highly integrated 2.5D and 3D eWLB solutions [27,28]. The development process is quite thorough and includes conformal shielding, system-level testing, high-density surface mount technology (SMT), and advanced packaging processes. High-density interconnects in low-loss package architectures are made possible by eWLB-based intermediary layers created by JCET, resulting in more effective heat dissipation and processing speed.

### 2.4. Integrated Fan-Out

Taiwan semiconductor manufacturing company (TSMC) created the integrated fan-out (INFO) for the first time in 2017. The FOWLP process is the foundation of the INFO, which completes the integration of many chips. With its compact dimensions and few connections, this package type prioritizes cost effectiveness. The INFO is appropriate for applications including the packing of high-performance computers (HPCs), high-frequency devices, and wireless chips. This form can provide space for numerous chips to be integrated at once [29]. A periodic electromagnetic band gap (EBG) structure in the form of an INFO package [30] is depicted in Figure 6. A cross-sectional view of this system is shown, along with micro-convex layer blocks, three different types of copper layers, and aluminum layers, and ultimately, the system is stacked on an SoC chip from top to bottom, with the metal layers connected by vias. The EBG is made up of a parallel-plate capacitor and two spiral-symmetric coupled inductors. Between Copper Layer 2 and Copper Layer 3, there are huge vias that connect the LC resonators. The system may deliver an excellent isolation environment with isolation levels up to 30 dB over a large frequency range (29.5 GHz).

### 2.5. Double-Sided Molded Ball Grid Array

The majority of double-sided molded ball grid array (DSMBGA) usage is in 5G networks. The integration of fundamental analog, digital, and radio frequency (RF) functions is necessary for 5G network packaging. In order to reduce conduction loss with extremely complex processing and high-quality processing materials, it is essential to merge multilayer board production with fine wiring. The co-simulation of the electromagnetic interference (EMI), components, and packaging must be completed at the same time. Amkor originally presented the DSMBGA packaging concept [31] and realized mass production of the product, as illustrated in Figure 7 for a schematic of a DSMBGA package, in order to achieve this high-quality 5G integration. The strip milling, molded underfill (MUF) method is mostly used for the top and bottom filling. This method divides the shielding region to reduce electromagnetic interference and places additional power amplification circuits and filters to improve signal integrity.

### 2.6. Embedded Multi-Chip Interconnect Bridge

Intel has made significant R&D investments in the embedded multi-chip interconnect bridge (EMIB) package. It is an organic-board-type package, unlike the other package forms, and lacks TSVs, which is based on a two-dimensional planar extension packaging technique. The whole EMIB package method is straightforward and high-yield since the EMIB is directly interconnected by silicon wafers for local high density [32]. This package offers good electrical qualities, a comprehensive shielding structure, and low crosstalk and loss. A very thin silicon bridge is placed in the top layer of the package system and connected to the chip pads of the substrate by vias in Figure 8, which is a schematic representation of the architecture of an EMIB [33]. The I/O’s SI and power PI of the entire system are essentially unaffected by the EMIB’s low and sparse pin count. When compared to typical package types, where the device must travel completely through the silicon intermediate layer, the EMIB’s electrical properties are much better.

Taken together, each of the above centralized SiP advanced packaging forms has its own characteristics, and the following Table 2 shows their comparison results [9].

The SiP forms are varied and show a convergence of numerous package advances so that they can also combine various advantages, whether it be the traditional wire bonding and flip chip or today’s fan-out and embedded die. To summarize, the main goals of such sophisticated packages are to increase power density, reduce size and line length, enhance system performance, and lower system loss. Additionally, SiP technology is now based more on the requirements of the application scenario. Computers will take into account the system’s processing speed; communication fields will take into account the signal’s stability during transmission; power modules will take into account the system’s ability to dissipate heat, etc.

## 3. Application of SiP Technology

The fast growth of technologies such as artificial intelligence and the Internet of Things has raised the demands on chips and systems since electronic information has now permeated every aspect of our life. The incomparable benefits such as the SiP’s tiny size, low power consumption, quick development time, and high adaptability are also present. As a result, the SiP is useful in an increasing number of fields. The application scenarios of SiP technology, such as wireless communication, power supply, central processing units/processors (CPUs), lasers, micro-electro-mechanical system (MEMS) sensors, etc., are outlined in this section.

### 3.1. Wireless Communication

SiP technology is now one of the key technologies in the field of wireless communication, where it was first used. The conventional packaging technology, which cannot satisfy the demands of high-performance devices, limits the ability of standard wireless systems to deal with the issues. By enabling the miniaturization and cost reduction of otherwise expensive and cumbersome devices at the system level, as well as in terms of dependability, SiP technology has emerged as a solution to these issues [34,35].

Millimeter wave refers to electromagnetic signal frequencies between 30 and 300 GHz, which broadly speaking comprises a variety of communication techniques such as radar, antennas, and sensors. The SiP of multi-layer/3D multi-chip modules is a more popular packaging technique [36], as it can combine the benefits of both thin-film and thick-film techniques with increased processing power and power quality [37]. With its adjustable size and shape in accordance with the actual application and perfect adaptability to the requirements of various components, as well as wiring, this upgraded method has low processing costs, high reliability, and high efficiency. As shown in Figure 9a,b for the five-layer and two-layer glass stacked structures, respectively, glass material has incomparable advantages for multilayer packaging of millimeter-wave antennas due to its good electrical properties and ability to be processed with high precision when compared to other materials [38]. The electrical connection between the glass and the peel is made using Cu and Sn. Layer-to-layer connections are made via a number of processes, including chemical gold plating, metal bonding, filling with plating, rewiring, seal ring sealing, and through glass vias (TGVs). With through holes for thick layers and blind holes for thin layers, the selection of holes is determined by the thickness of the layer. During testing, it was discovered that the antenna formed of a glass wafer has a more improved gain and is more stable when dealing with complex and varied antenna designs.

A ceramic substrate is another excellent option in addition to a glass substrate. The first one is a high-temperature co-fired ceramic (HTCC), which buries the vertically inflated linked waveguide and muffles the mixer’s harmonic output by making use of strong mechanical strength and stable chemical strength. Radar arrays can use this technology because of its extraordinarily high transmission efficiency and guarantee of function realization [39]. The next application is a low-temperature co-fired ceramic (LTCC), which is for high-frequency communication. It was first proposed for semiconductor transceivers above 100 GHz, taking the mechanical defects of the LTCC into consideration [40]. The viability of the scheme was demonstrated after the completion of the robust design and standard assembly. The benefits of this material substrate for the SiP will be more obvious if the “hardware” weakness of the LTCC can be fixed.

### 3.2. Power Modules

The majority of power electronics use power modules to supply power. System-level packaging can resolve the seeming contradiction between the high power of power modules and the downsizing of size and weight [41]. The following difficulties should be taken into account when using the current SiP technology for power modules [42,43]: (1) realizing effective thermal control to enable efficient heat dissipation; (2) minimizing the parasitic effects brought on by the connection of various devices and enhancing the electrical characteristics; (3) possessing a certain capacity for external loads; (4) balancing the effects of the size, weight, and power loss. Since frequent vibration work environments are common for high-power modules in the automotive, aerospace, and other industries, the design of the package shape primarily considers mechanical stability. The assembled three-dimensional structure of a power module, including a driver chip, MOSFETs, inductors, and discrete resistors and capacitors, is schematically depicted in Figure 10 [44]. There are two layers: the upper layer includes two MOSFETs, as well as extra resistors and capacitors, while the lower layer has four huge inductors in addition to the upper board’s same components. Through a connecting board containing solder balls, the electrical connection between the two layers is made. The shapes and spacing of the solder balls can be modified to minimize the maximum stress and increase the dependability of the system through the use of finite-element analysis and actual solder ball measurements.

### 3.3. Computers

Computers are innovating towards low power consumption, low cost, and high computational throughput. The idea of developing advanced computer processors and memories has shifted to SiP technology with smaller heterogeneous systems [45]. For instance, the European Microsystem Package 32-Bit Computer Module Unit [46] has a total mass of only 150 g and a power consumption of just 12 W while integrating a central processor, static data memory, dynamic data memory, and bus and clock resources. SiP technology has also been applied by a number of researchers to computer processor packaging. As shown in Figure 11, a silicon adapter plate is mounted on the package substrate using the CoWoS technology [47], which mounts multiple chips onto a silicon intermediate layer and interconnects them with high-density wiring. The CoWoS intermediate layer is attached to a single substrate to complete the package, which consists of two chips in total. The two chips are connected to the intermediate layer, where they are positioned below via bumps. The number of bumps in the intermediate layer is twice as many as those in the package substrate, giving the chips more places to be wired. The benefits of CoWoS are highlighted because the two chips are connected directly by a low-voltage-in-package-interconnect (LIPINCON), and the data processing speed of this package can be increased to 8 Gb/s.

### 3.4. MEMS Sensors

The three pillars of information technology are MEMS sensors, communication technology, and computer technology. Conventional sensors are heavy and bulky, so they have high energy consumption. MEMS sensors are increasingly becoming more common since portable and miniature sensor devices are expected. SiP technology is one of the greatest options for MEMS sensor innovation because it is crucial for shrinking and lightweighting [48,49]. Figure 12 depicts the schematic architecture of a gas sensor [8], with the bottom substrate attached to the sensor by lead bonding and the top substrate fit with tube cores for temperature and humidity. A TSV loaded with conductive silver adhesive is used to electrically connect the substrate’s bottom and top layers, while thermally conductive silicone grease is used to affix the remaining layers. The complete package system is far smaller than conventional printed circuit boards, making it more useful for evaluating specific situations with stricter volume restrictions. In addition to gas sensors, devices such as temperature sensors and ultrasonic rangefinders [50,51] can reduce the complexity of measurement while also increasing measurement accuracy by converting heavy equipment into small sensors.

### 3.5. Lasers

We always ask lasers to operate at higher luminous powers, wider arrays, and faster response times while still fitting them into small spaces [52]. The breakthrough in SiP technology for laser packaging is more necessary than ever. A small laser or other optical components are linked to the chip to create a complete system that is then connected to an electrical pin and soldered to a PCB [53,54,55]. This kind is different from the packaging for the numerous application cases outlined in the preceding section [56]. Figure 13 depicts an integrated package form of a femtosecond laser [57], with a silicon photonic chip on the left connected to 12 multicore fibers through 84 connecting channels in the center, while glass serves as the intermediary layer. The proximity fiber encapsulation is also schematically depicted in Figure 12, and it is created by laser irradiation and chemical etching to precisely match the fiber’s hole diameter. The package also switches from a challenging five-axis connection to a simple single-axis connection thanks to the high accuracy of both methods. With its ability to properly balance loss, channel density, size, crosstalk, and other parameters, as well as its ability to reduce the burden associated with device assembly, these design and processing serve as a model for various coupling packages of micro lasers and precision photonic devices.

As can be seen, the SiP is applied to numerous facets of our lives, further demonstrating the importance of this packaging type. However, the implementation of new processes and structures also introduces a number of reliability issues. If not detected and solved in time, it will make the system work abnormally and then lead to equipment failure or even major accidents.

## 4. SiP Reliability Analysis and Testing

All designed package structures must be analyzed for reliability in order to have application value. The reliability of the SiP refers to the likelihood that a product will fail or fail inside a specific phase and is a crucial metric for determining if the packaged product can function steadily over the course of a given life cycle. If we look at the failure form in terms of the research of package dependability, possible issues include warpage, chip cracking, delamination, toughness fracture, plastic deformation, and many more. If we break them down further, there may be hundreds of them, which makes it difficult to describe and raises the cost and testing effort. The dependability issue, however, may be broken down into three categories if we look for the source: thermal management, mechanical stress, and electrical properties. This way will make it simple to study reliability issues in an understandable manner and uncover the root reasons quickly based on the packaging failure form. In fact, there are many potential failure modes in the system, such as stress fracture, high-temperature deformation, high-temperature deterioration, open circuit, short circuit, line impedance mismatch, electromagnetic interference, and many other categories. Reliability mainly revolves around the above potential failure modes. The analysis and testing of temperature, mechanical stress, and electrical system-level reliability issues are summarized in this section.

### 4.1. Thermal Management

Since no electronic device can now operate at 100% conversion efficiency, some electrical energy must be transformed into heat. With the unique advantages of small size and high density, the SiP helps to achieve system integration. Due to the stacked arrangement of multiple chips, the heat density inside the SiP is higher than that of discrete devices, and the heat transfer paths are further restricted in a limited and narrow space. According to the heat transfer equation, the temperature rise problem caused by the SiP’s special structure is more severe than that of packages such as TSOP.
(1)$$Q=kA\frac{\Delta T}{\Delta x}$$
where Q, k, and A are heat, the heat transfer coefficient, and the heat transfer area, respectively. ΔT and Δx are the changes of temperature and distance, respectively.

The heat dissipation of the entire system will be impacted and the internal temperature will rise if the thermal conductivity of the encapsulation system is not resolved. The mechanical stress will change irregularly as a result of excessive differences in the coefficients of thermal expansion (CTEs) of adjacent materials. In addition, as temperature rises due to ongoing heat buildup, component performance is also impacted, which in turn affects the system’s overall electrical properties. As a result, the majority of reliability issues are caused by temperature issues, which necessitate thermal characterization analysis.

#### 4.1.1. Thermal Characterization Methods

Currently, thermal characterization can be accomplished in two steps: simulation and measurement. A few popular techniques are given below:Finite-element analysis (FEA):

In terms of thermal analysis, finite-element analysis has advanced considerably. There are other simulation programs besides ANSYS, Comsol, Flowtherm, etc., that can be utilized for the thermal characterization of the SiP, whether it is a 2D or 3D structure. According to the survey, the SiP applications of FEA are more likely to restore the functioning scenario of the SiP [58,59,60,61], such as by adding suitable temperature cycles, simulating the architecture made up of silicon vias and bumps, modeling reflow soldering, etc. Modeling is a must for finite-element analysis, but the 3D kind of SiP is complicated, and if the simulation’s accuracy is to be ensured, the computational time will undoubtedly grow and the efficiency will be significantly decreased. The global multi-structure 3D model is reduced to a 2D structure with separate structural coupling connections [62] when a quasi-3D structure is utilized, which can meet the accuracy requirements and speed up FEA computations. In general, FEA methods still address modeling problems, especially concentrating on the cross-scale and cross-dimensional aspects of 3D models.

Thermal imaging technology:

Thermography is the act of turning thermal radiation into an electrical signal that can then be amplified and processed to create a thermal image that shows the temperature distribution throughout an object’s surface. Thermography can be used to measure the local temperature of microstructures such as the SiP or MEMS [63], with the benefit of being quick and easy to use, but the drawback of only being able to assess surface temperatures.

Phase-locked infrared thermal localization:

Phase-locked infrared thermal localization overcomes the limitations of two-dimensional measurement and may find deep hot spots to address the inadequacies of thermal imaging methods [64]. This method is an excellent complement to thermal analysis for 3D designs and employs computer simulations to find flaws inside 3D stacked structures via surface imaging. The drawback is that the phase-locked infrared localization algorithm is more complicated and takes a while to compute.

Thermal resistance matrix method:

If a single heat source calculation is still performed, SiP technology’s multi-chip stacking will surely result in several heat sources throughout the entire system, which will result in a considerable difference between the simulation results and the actual ones. In order to quantify the coupling and mutual influence between various heat sources with high reliability in thermal characterization, the thermal resistance matrix of the 3D stacked package must be created [65]. In general, the accuracy of this method improves as more chip parameters (temperature, structure, material, and electro-thermal characteristics) are collected.

Scanning:

The scanning method is another adapted concept used to describe the SiP’s thermal characterization. With high testing efficiency and accuracy, X-rays can be utilized to scan the entire packaging system and complete the structural and functional testing, which is extremely valuable for high-precision packing equipment.

#### 4.1.2. Thermal Characterization Objects

Presently, researchers studying thermal characterization are mostly concentrating on substrates and ball grid array (BGA) solder connections, both of which are important structures that have an impact on the SiP’s thermal dependability.

A guarantee for the appropriate operation of the entire system is if thermal issues can be properly resolved because the substrate serves as a crucial carrier for the electrical connectivity and physical support of the SiP. At this point, simulating the substrate is necessary both alone and in conjunction with connected components. Computational fluid dynamics (CFD), for instance, is utilized to examine the interconnection of the silicon chip and LTCC substrate and the evolution of the bonding process between them [66]. According to research on the thermal properties of three different package types—2.5D package, fan-out chip on substrate (FOCoS) chip-first, and FOCoS chip-last—the junction temperature of the FOCoS type is lower than that of the 2.5D package, meaning that the substrate material and the design of the structure both have effects on the thermal properties [67]. The two FOCoSs’ shared grinding operation, which uniformly thins the material between the heat sink and the package and, so, ensures overall thermal efficiency, is the primary cause of this.

BGAs meet the requirements of the SiP because they feature a large number of pins, a high yield, and good electrical characteristics. Yet, in order to link devices to the PCB, BGAs are inescapably exposed to heat-related stressors, which might in turn cause reliability issues. The present research strategy for BGAs’ thermal issues is straightforward: typically, the architectural model of the system and the parameter details of each device are first established, and then, simulations in finite-element software are performed to obtain the related thermal results. Following completion, measurements are taken using machinery to confirm them. The difference between the model utilized and the actual computed findings was less than 2 °C in the low-temperature range and less than 4.5 °C in the high-temperature range, according to control tests carried out by certain researchers using a thermal imager [68]. The accuracy of the finite-element solution has increased, and the modeling parameters have been continuously subdivided and supplemented. As a consequence, the thermal simulation results of the BGA closely match the actual values, which paves the way for further BGA optimization.

### 4.2. Mechanical Stress

Another important factor in determining the dependability of SiPs is the examination of mechanical stress. In system-level packages, the materials used for the chips, substrates, leads, and solder connections vary. Different materials will shrink or expand in accordance with their unique qualities when subjected to influences such as temperature change or external impact, leading to specific deformation and consequent mechanical stress issues. As an illustration, we give the formula for the CTE:(2)$$\alpha =\frac{\Delta l}{l\times \Delta T}$$
where α, l, and Δl are the CTE, length, and length change, respectively. When two materials are heated, their CTEs differ, the strain will also be noticeably inconsistent, and a certain amount of stress will be produced, which falls under the category of mechanical stress. Loss failure and overstress failure are the two primary types of failure brought on by mechanical stress. The former refers to the buildup of low stress that is continuously applied over an extended period of time and results in component loss, leading to system failure; the latter refers to failure brought on by a great stress that exceeds what the components can withstand during a specific event or moment. More focus has been placed on loss failure because, as was previously mentioned, a significant portion of the mechanical stress is caused by an imbalance in the CTEs of various materials as a result of the temperature rise. Therefore, it is more advantageous to investigate the specific causes of failure. Moreover, loss failure is more valuable to examine because it occurs frequently in all packing systems, whereas overstress failure is more like an unforeseen, uncontrollable catastrophe.

#### 4.2.1. Stress Measurement Methods

Lossy testing and nondestructive testing are two different ways to evaluate stress. The latter is non-destructive, flexible, reliable, and dynamic, which is better suited for system-level packaging, which requires meticulous and exacting testing. The following non-destructive testing techniques are widely applied:Diffraction method:

The diffraction method uses the properties of X-rays, which diffract when traversing materials such as polycrystalline metal surfaces. In the event of stress, the resulting lattice distortion means the stress of the relevant material in accordance with Hooke’s and Bragg’s laws. This technique is well suited for use in 3D integrated measurements and can quantify both macroscopic and microscopic residual stresses [69].

Indentation method:

Impact loading is used in the indentation method to make an indentation in the center of the strain flower, and resistive strain gauges serve as the sensitive element for measurement. A strain gauge records the change in strain increment in the indentation area and uses that information to calculate the stress value. To examine the TVS stress analysis of the SiP, the indentation method has recently been enhanced to the nanoscale level with extremely high stress resolution and displacement resolution [70].

Curvature method:

The level of constructive stress indicates the magnitude of strain at the material interface of each layer, while curvature is a quantitative measure that reflects the degree of curvature of a curve or surface. At the microscale, residual stresses normal to the material interface are calculated using the curvature method, and a thin film is chosen as the measurement object. If chosen properly, this measurement method is simple, effective, and highly accurate [71]. However, if the chosen film cannot replace the original stress scene, the measurement method may have some flaws.

Blind hole method:

The blind hole technique entails making a tiny hole in the region that needs to be measured. The stress field here changes as the stress close to the hole is released, and by measuring the new value, the original stress value may be deduced [72]. Almost all types of materials in SiPs can be used with this technology, but the resolution of the stress measurement might not be as good as with the above methods.

#### 4.2.2. Mechanical Stress Analysis

Two ideas can be used to analyze the mechanical stress reliability of SiPs: one from the study’s subject and the other from typical failure types.

The SiP structures that are susceptible to mechanical stress are solder joints, bumps, and package boards. Bumps are similar to solder joints, which play the role of connecting the substrate and device. Due to the mismatch of the thermal expansion coefficients of different materials, this will lead to both being susceptible to mechanical stress. After a period of accumulation, the continuous mechanical stress brought about by the temperature increase will reduce the working life of the device. Thermal cycling tests are typically performed to create stress or strain maps of the solder joint array for the tropicalization stress problem, which is then followed by finite-element analysis [73]. In general, the most-stressed solder joints in embedded SiPs are found in the middle of the package, whereas they are found towards the margins of the package in the standard SiP. In addition to thermal cycling, the impact of vibration on the solder joints or bumps has also been investigated using a type of simulated random vibration [41]. To determine the associated danger spots, a secondary simulation of the calibrated model is run with the model’s parameters changed. To determine how long a solder joint will last, some of them can be incorporated into a life prediction model [74]. Traditional testing techniques, including temperature shock tests, temperature and humidity deviation tests, etc., are used to test the stability of package boards—typically SiPs with a laminated chip embedding. At high temperatures, specific types of stress manifestations, such as those brought on by local temperature gradients, can lead to higher expansion of the packaging system, which needs intermittent service life testing. Conventional stress testing techniques typically focus on the impact of a single stress, but in reality, failures are frequently caused by the coupling of pressures brought on by a number of different variables. Because of this, some people have adopted the strategy of combining stresses to create novel stress measurements, such as investigating the effects of both thermal cycling and humidity, i.e., the superposition of both forms of stress on the filler material [75], which frequently more accurately depicts the actual package stress situation.

Warpage, deformation, and other problems are the principal mechanical stress-related failures of the SiP. The amount of an arbitrary surface’s out-of-plane displacement is measured using the term “warpage”. It is essential to reduce the warpage phenomena of SiPs since wafer-level packaging technology is making the package material thinner and thinner [76]. Figure 14 displays a warpage demonstration for an epoxy molding compound (EMC), with two routes, AB and CD, which may calculate the warpage, as well as the maximum and minimum warpage axes [77]. Solving the CTE of an unknown structure is a necessary step in the analysis of the failure form. In order to determine the CTE before and after each process, researchers have adopted a new idea of measuring the CTE by using the thermal profile of a similar process to analyze the thermomechanics. This method is known as process thermomechanical analysis (pTMA). It also makes it possible to ascertain whether or not the changes brought on by temperature are reversible. As a result, pTMA seems to be a reliable analysis technique for CTE-related dynamic mechanical stress analysis. The primary cause of the deformation is the stress shock, which 3D SiPs still need to identify. The sensors that can measure stress and vibration in the folded region of the face to face are often present in the more widely utilized cubic SiPs in recent years. By incorporating the system into a package and simulating a typical working situation with shock or vibration while using the embedded sensors for evaluation, the reliability of the cube package has been confirmed [78].

### 4.3. Electrical Properties

The SiP’s reliability concerns ultimately manifest as problems with the electrical properties as a result of the thermal and mechanical stress problems. Examining the electrical performance is another technique to determine the SiP’s dependability or to see if it functions as intended. The electromagnetic environment of the entire package system is extremely complex because of the increasing density of SiPs, and the interconnection channels’ effects on the signal are becoming more and more important. It is crucial to take into account a number of factors while evaluating electrical performance, including EMI, SI, and PI.

#### 4.3.1. Electrical Properties Measurement Methods

The following can be used to categorize the analysis and testing of electrical properties:Model construction method:

In order to understand and solve the electrical parameters, the model-building process typically includes replacing a complex packaged system with a suitable circuit model. The electromagnetic theory of Maxwell and fundamental circuit models are some of the modeling foundations. The majority of electrical-performance-related aspects can be addressed using this methodology. A double-sided component placement (DSCP) model based on RLC circuits can be created for PI [79]. In order to assess the performance of the entire power distribution network (PDN), the DSCP model includes the impedance analysis from the power supply to the motherboard and from the regulator to the IC, as illustrated in Figure 15. Some researchers have tested electromagnetic interference using a progressive RF optimization method [80]. A reference for the precise analysis and optimization of the near-field coupling between lines can be found in the use of dual-interconnected lines as a model to study the characteristics of near-field coupling and the establishment of an equivalent aggregate circuit model to specify the complex model with devices. In order to solve the circuit’s connectivity security, a novel measuring concept has been put out [81]. The common method of measuring open-/short-circuit problems in the RDL is given a model, to which a simplified diagram of the measurement system, including insulation resistance, capacitance, converter gain, etc., is added. By analyzing the two DC signals given by the former level circuit, it is possible to determine whether there is an open or short circuit. The whole process can be performed by using mixing technology and a few passive devices, which greatly simplifies the testing steps.

System integration method:

Using external devices for testing still presents some challenges because the SiP is a sophisticated system. Direct integration on the SiP has been suggested by researchers, including the joint test action group (JTAG) [82]. The JTAG can be tested directly through the SoC in the multiplexed system for the testing of SiP devices. The transient dose rate effect (TDRE) of the SiP [83], which results in the combination of electrons and holes, producing photocurrents, which subsequently interfere with the circuit, is treated similarly. The response method involves programming and recording the characteristics and locations of abnormal signals in the circuit using a field-programmable gate array (FPGA), which is already built into the SiP.

Numerical calculation method:

A complex computational issue, the analysis of the electrical properties of SiPs calls for advances in numerical computation, increasing the effectiveness of data solving [84]. For instance, there are billions of unknown quantities in the high-performance JEMS-CDS software built on the JAUMIN platform that need to be resolved. The solution speed can be significantly increased by using the orthogonal matrix and region decomposition preprocessing approach. For issues such as EMI and SI, when the solution volume is enormous and the computing domain is divided, this kind of method is particularly useful.

Scanning method:

The scanning technique is also a more typical way to examine the package in question’s electrical characteristics. For electromagnetic interference issues, near-field scanning can be performed to determine whether the SiP’s shielding function is efficient [85]. The SiP substrate’s dielectric constant has a significant effect on the signal in high-frequency scenarios in terms of SI, and two-dimensional scanning can be utilized to test the substrate’s dielectric constant as well [86].

#### 4.3.2. Electrical Properties Objects

The fourth-generation of the interconnect technique, the TSV, is the primary structure for achieving vertically oriented connections in SiPs [87]. To confirm the reliability of more TSV structures as they are created, related electrical performance testing tools are required. For instance, the RLCG parametric analytical formulation equivalent circuit model was developed to study the dependability of strip TSV structures, and simulations with ADS and HFSS were used to compare and validate the model’s viability. The impact of the size of the strip TSV on the signal integrity was examined [88] in order to emphasize the benefits of the strip structure. As shown in Figure 16a, the longer the long side, the smaller the cross-sectional area, which can make the leakage current decrease and, thus, reduce the insertion loss, given that the sum of the circle diameter and the long side of the bar TSV is constant. As seen in Figure 16b, using a strip TSV with a long-edge feature can enhance the system’s SI. Furthermore, specific cases have been investigated using millimeter-wave test structures to examine the transmission characteristics of a single TSV, a dual-redundant TSV, and a quad-redundant TSV in the presence of specific system faults [89]. The HFSS software has been used for simulations to show that the redundant class of TSVs can operate autonomously at high frequencies (44 GHz), even in the presence of fault issues. In the experimental portion, the RF redundant TSV test apparatus was constructed, and the insertion loss at 40 GHz for the dual-redundant and quad-redundant TSVs, respectively, was measured to be 0.19 dB and 0.46 dB, with 0.22 dB for the single TSV. The ability to handle faults is improved by the redundant TSVs, despite the fact that the insertion loss grows as the number of TSVs does.

After mastering the reliability analysis of the SiP in three aspects: thermal management, mechanical stress, and electrical properties, we can try to propose an optimized solution based on the causes of failure.

## 5. SiP Reliability Optimization Solution

The optimization objectives in this section can also be divided into three categories: the goal of thermal management is to limit the maximum temperature to less than 120 °C during a period of normal operation; the goal of mechanical stress management is to reduce the maximum mechanical stress value as much as possible; electrical performance will be further divided into a voltage drop less than 5% of the maximum supply voltage, a current density less than 30 A/mm^2^, an SI important signal return loss less than −40 dB, and EMI indicators required to be less than 1 v/m.

### 5.1. Thermal

Three basic methods have been proposed to address the SiP thermal issue: the addition of a heat sink; the use of materials with high thermal conductivity; the optimization of the SiP structure.

#### 5.1.1. Adding Heat Sinks

The high-density system must contain a heat dissipation part, and the temperature rise caused by the operation of high-power devices is solved by heat sinks or other heat dissipation methods [90]. Monolithic microfluidic cooling, which is a low thermal resistance silicon fin engraved on a small chip, is an example of a cooling solution applied to small sizes [91]. Figure 17 depicts a schematic of monolithic microfluidic cooling, which places microneedle fin heat sinks on the chip and adds manifolds around them to facilitate coolant flow and transfer. This small heat sink, which has a thermal resistance of 0.07 °C/W and can maintain a constant temperature of 30 °C in an FPGA system with a power of 107 W, is more practical in space-dense systems than huge, bulky heat sinks. Some researchers have compared the shapes of the same type of radiators and found that tree-fin-type radiators are superior to parallel fins and interlaced fins [92]. Since the tree fins allow the fluid between adjacent channels to flow through one channel first and then separate into two channels, it is equivalent to an equalizing effect of heat. This method can reduce the temperature difference inside the system, prevent the problem of uneven heating of the device, and facilitate the overall heat dissipation.

#### 5.1.2. Using High Thermal Conductivity Materials

The solution to high temperatures is to conduct heat out quickly, so the use of highly thermally conductive materials is also a viable method. In order to function properly, pulse sequencers functioning in hot conditions urgently need to channel much heat away from them. To provide good thermal conductivity, an ultra-thin conductive adhesive is typically applied between the substrate and the SiP’s base. Decoupling capacitors, which also have a high thermal conductivity, are one of the COG materials [93]. Inadequate thermal management can cause deformation and failure in solder joints [94]; to increase reliability, filler adhesive materials can be used, such as epoxy resins with additional components such as boron nitride and aluminum trioxide, which have strong temperature resistance and thermal conductivity. According to the experimental findings, the inclusion of filler glue increased the fatigue life of solder junctions by a factor of three [95].

#### 5.1.3. Optimizing SiP Structure

To overcome the problem of thermal reliability, one can study the thermal reliability from the design structure of the SiP itself in addition to the typical addition of heat sink and thermal conductive materials. In the SiP system, there will be numerous integrated chips, which means numerous heat sources. The thermal reliability of SiPs can be increased through logical layout and optimization at the two-dimensional plane or height level [96]. The best outcome, using the optimization of dual-hot spots as an example, is to be able to logically arrange the two heat-generating chips so that the heat dispersion between each layer is very consistent. By building a heat transfer model of the 3D temperature field, a neural network is used to obtain the intrinsic connection between the hot spot map and the different chip placement positions. In order to find the ideal hot spot location, a multi-objective genetic algorithm can be used to solve the objective functions of the maximum temperature and the uniformity of the temperature distribution. Figure 18 illustrates the general symmetry of the temperature distribution, which strikes a reasonable balance between the maximum temperature and the even dispersion of heat [97]. Vias also improve the thermal performance, so adding grounded vias around the stacked chips is a good option. The heat is transferred to the metal layer through the vias and then dissipated using air cooling or convection [98]. The thermal dependability of the packing system can also be impacted by factors including the filling material, diameter, thickness, and spacing of the TVSs [99]. After simulation and experiment, it was discovered that, on the one hand, we can increase the thickness of the TVS high-thermal-conductivity filler material and decrease the TSV spacing; on the other hand, we should decrease the number of TSV chips used and increase the chip size, properly.

### 5.2. Mechanical Stress

In the vast majority of cases, mechanical stress and thermal management work together to cause failures such as warpage, plastic deformation, and fatigue fracture. Mechanical stress between different materials is caused by the different CTEs due to the rise in temperature caused by the operation of the system. Therefore, the above-mentioned increase in thermal reliability partially resolves the issue of mechanical stress. The solder pins and lead bonding of SiP technology are also the more serious reliability difficulties, in addition to the thermal–mechanical reliability challenges.

Aluminum nitride has a high thermal conductivity and a similar CTE to the chip, making it an ideal material for making substrates. The reliability issues that need attention are mainly focused on the strength of the connection between the aluminum nitride substrate and the metal pins. Figure 19 depicts a model diagram of the established stress test, which can be used to investigate the variables influencing the solder strength of the pins [100]. The substrate is aluminum nitride, and the two lead types can be investigated for vertical tension and peel force, respectively. The vertical lead is a nail head lead soldered to a circular pad, and the horizontal lead is a flat lead soldered to a square pad. It was discovered through testing that increasing the thickness of the nickel plate layer and the quantity of brazing material used for soldering would result in a reduction in the strength of the pin soldering. The strength of the pins is also influenced by the brazing’s structure. By comparison, the strength of the soldered pins can be significantly increased by using a soldering structure with bent pins.

Copper-to-copper/silver bonding is one of the key technologies in the packaging section of SiP technology, which depends on the bonding technology to achieve chip interconnects [101]. The bonding strength can be significantly increased by creating a copper nitride coating on the copper surface using a two-step plasma process, which can produce low-temperature copper bonding and a shear strength of up to 30 MPa [102]. Figure 20 depicts a schematic diagram of the two-step plasma treatment [103]. The silicon wafer is first covered with a copper layer, and the deposition method’s addition of a reinforced copper film with an argon gas stream is the first plasma treatment phase. When the surface impurities are cleaned up, the second plasma treatment phase is carried out, utilizing nitrogen gas to stop the copper coating from oxidizing in the presence of oxygen. This process improves the mechanical durability of the SiP by providing the bonded copper with outstanding passivation capability, stability, and high shear strength.

### 5.3. Electrical Properties

Electrical performance has a fairly wide range, so dependability issues should be taken into account with the scenario’s particular application. For instance, the SI is particularly crucial for high-frequency signal transmission; for power modules or DC–DC, the general focus is on PI; EMI is the main area of examination when numerous independent modules are combined to provide system operation.

#### 5.3.1. Signal Integrity

SI issues can easily occur during signal transmission if the transmission line’s impedance is discontinuous, which can easily lead to reflection phenomena. Transmission lines, vias, and solder joints on the substrate make up the whole signal transmission path in the SiP, and an evident impedance discontinuity is present. The SiP structure must be tuned to make impedance matching in order to increase SI, which includes choosing substrates with strong RF properties and a low dielectric constant as the adapter board, optimizing the size and arrangement of solder connections [104]. Impedance matching can also be achieved in the machining process, such as the use of shaped ramp interconnection techniques, which allow impedance matching in the K and W bands [105]. As an illustration, Figure 21 displays a schematic of a two-dimensional cross-section of a ramp interconnect that was inkjet-printed on a MEGTRON 6 substrate [106]. The tube core is put in place; the substrate is initially bonded with ink, and then, the attachment is finished with ultraviolet light. Immediately after, four layers of ink are printed, and ultraviolet light is still used to cure the deposition ramp that connects the substrate to the top of the die. The full ramp can be used for multi-layer inkjet printing and has a maximum slope of 35 degrees. The insertion loss of each ramp connector after this procedure was empirically confirmed to be 0.45 dB/mm, which is entirely acceptable in the high-frequency band.

#### 5.3.2. Power Integrity

The power supply network is made up of decoupling capacitors, the ground, and power. Parasitic effects are easily present in high-speed or high-frequency environments, causing voltage drop and power supply noise, which can further impair the chip’s power supply or the transmission of other crucial signals. Increasing the width of the power supply alignment, rationalizing the device placement, minimizing the parasitic effects, and shortening the distance between the power and ground are typical ways to increase PI. A more representative case for improving PI in SiPs is in the package design of inductive regulators. This is usually performed by placing the regulator as close to the load as possible so that the power management loop is shorter and the parasitic effects are minimal, improving power integrity [107,108]. Inductive regulators can be packaged using an embedded planar manufacturing technique. A printed circuit board is filled with a composite magnetic material that is compatible with the board-level material and is etched in accordance with the inductor’s size. The self-resonant frequency can be raised by 3.4% with this technique, and the quality factor can be raised by 20%. A TSV-based electromagnetic bandgap structure has been suggested by other researchers as a method of assisting the PDN in achieving broadband noise suppression. In comparison to the standard EBG, the EBG with the integrated TSV device may be employed in a gated PDN with a maximum rejection bandgap of up to 13.22 GHz. The bandgap can be altered by altering the structural parameters for various scenarios, which is stable and useful [109].

#### 5.3.3. Electromagnetic Interference

The EMI problem stems from the fact that the interference source passes through a coupling path and transfers energy to other sensitive components. There are three forms of implementation: conduction, radiation, and near-field coupling. It is usually overcome by the rational layout of the power supply and ground layer, partitioning of the same type of devices, and increasing the shielding structure. The more widely used research has concentrated on the topology and addressed the present EMI problem for the integrated package. For instance, when applied to the single-branch LLC resonant converter’s high-voltage power supply architecture, the performance of the converter declined as the number of turns ratio increased. The secondary can also be linked to the relevant load if the configuration is altered to a scalable distributed LLC topology by cascading multiple primaries and secondaries in order to eliminate the voltage drop on each branch’s primary winding. With this method, the turns ratio can be reduced to achieve a high buck ratio conversion by increasing the number. The distributed LLC’s maximum radiative power is 24 dBm, and its maximum electric field strength is only 210 V/m, or 28% of a single branch, according to an EMI near-field simulation, which substantially resolves the EMI problem [110]. Additionally, it is worthwhile to investigate the package voltage regulator (VR) on the SiP placement, as shown in Figure 21. Both structures contain four components: an IC, a VR, a SiP system, and a decoupling capacitor at the bottom; the red and black lines denote the 12 V and 0.8 V power supply networks, respectively [111]. The only difference is that Figure 22a places the VR on the top level and Figure 22b places the VR on the bottom level. The former offers additional layout benefits because it can both minimize the direct connection distance to the IC and free up space at the bottom for the standard arrangement of BGA pins. The bottom configuration has relatively substantial EMI problems, according to simulation findings from ANSYS SIwave, because the VR is closer to the BGA, causing a larger direct power network from the linked power pins. This ideal layout method on the topology of independent devices can successfully address the system’s EMI problems.

## 6. Conclusions

Future advancements in circuit technology will make them more streamlined and integrated, raising the bar for packaging technology. SiP technology is a good fit for these requirements in terms of definition and practical use. More importantly, the current competitive pressure in the packaging industry is such that companies are striving to be able to achieve high-volume and high-reliability production at a low cost. A known good die (KGD) for multi-chip assemblies is a good way to do this and, to a certain extent, also provides favorable conditions for SiP innovation. In this review, the history and advancement of SiP technology were discussed in relation to the current broader context, and the features and benefits of SiP technology were compared and analyzed. After investigating the advanced SiP technology of a large number of well-known companies, we summarized the fields of the SiP’s applications, including communication, lasers, MEMS, power supply, etc.

SiP technology advancements also result in issues with package dependability. This review outlined the techniques for assessing the SiP’s reliability and, in addition, focused on the elements impacting reliability. The reliability difficulties were broken down into three categories: thermal management, mechanical stress, and electrical properties. Finally, it looked at how to increase package dependability from the three perspectives, discussed how to improve SiP designs, and provided practical examples that might serve as appropriate benchmarks for future SiP designs.

SiP research will keep concentrating on developing novel package structures and enhancing package reliability for some time to come. Both a challenging research question and the packaging industry’s future development trend revolve around how to maximize performance under the constraints of volume and weight limitations. If reliability can be further ensured, the SiP will have a broader development prospect in various industrial fields.

## Figures and Tables

**Figure 1 micromachines-14-01149-f001:**
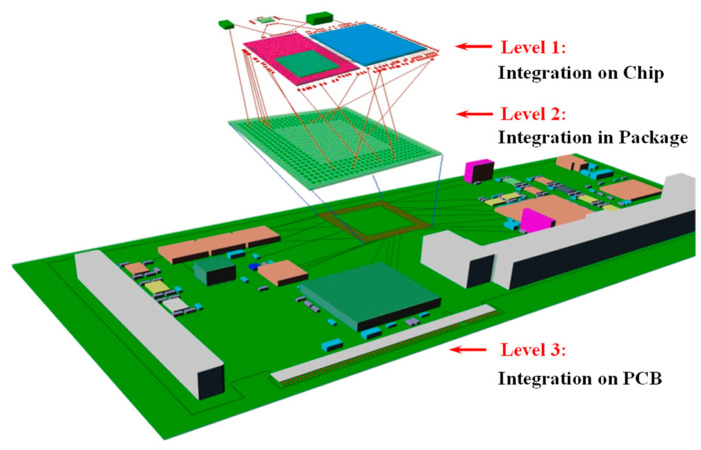
Level classification of electronic integration.

**Figure 2 micromachines-14-01149-f002:**
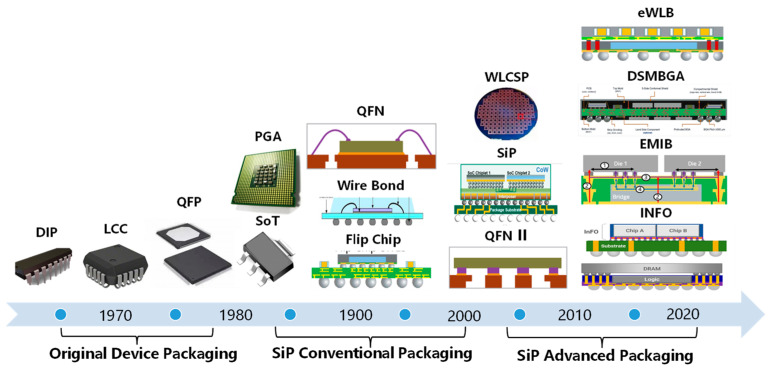
Trends in SiP.

**Figure 3 micromachines-14-01149-f003:**
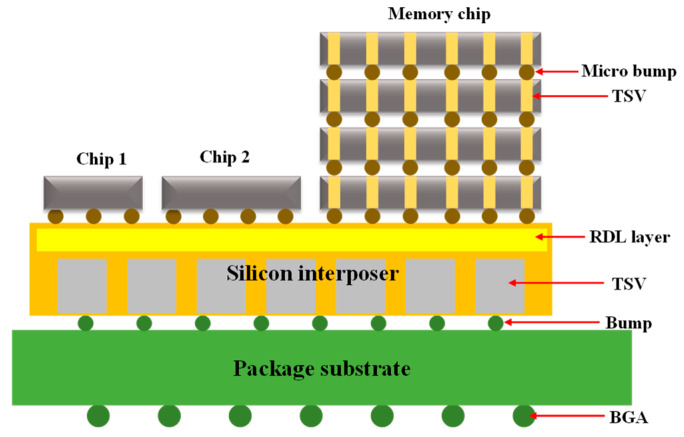
Typical SiP structure schematic.

**Figure 4 micromachines-14-01149-f004:**
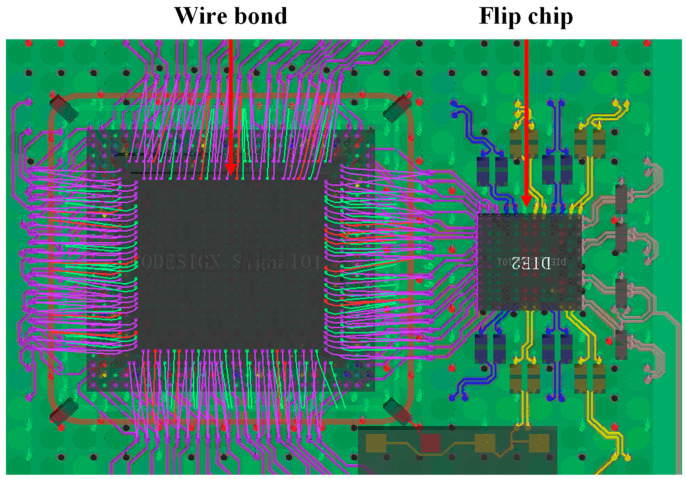
SiP interconnection form: wire bond and flip chip.

**Figure 5 micromachines-14-01149-f005:**
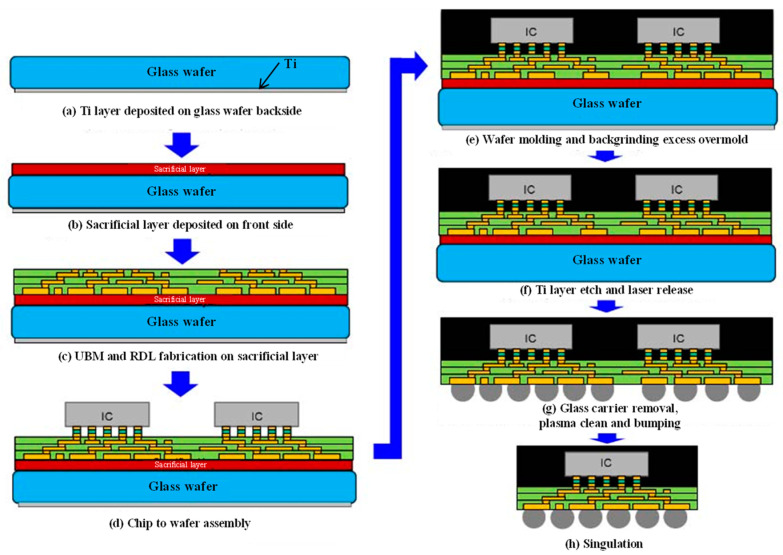
FOWLP process integration flow.

**Figure 6 micromachines-14-01149-f006:**
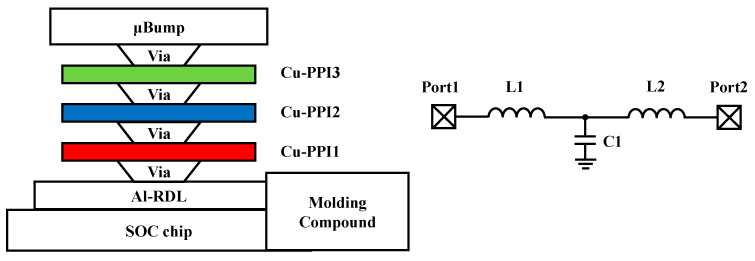
Schematic of EBG in INFO package.

**Figure 7 micromachines-14-01149-f007:**
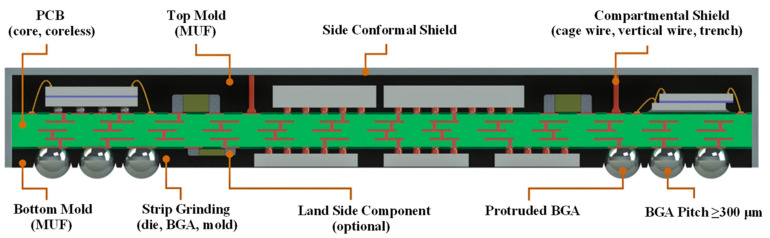
Schematic of DSMBGA package concept.

**Figure 8 micromachines-14-01149-f008:**
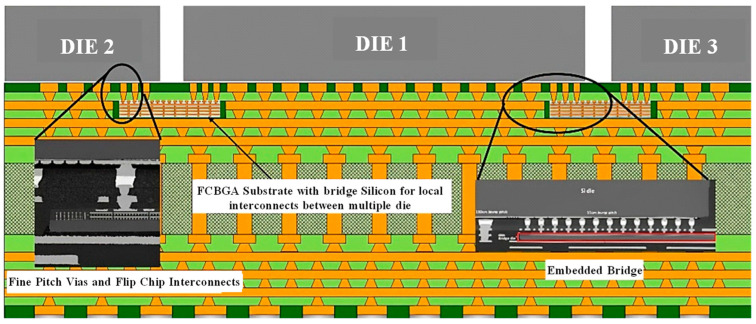
Schematic of EMIB system architecture.

**Figure 9 micromachines-14-01149-f009:**
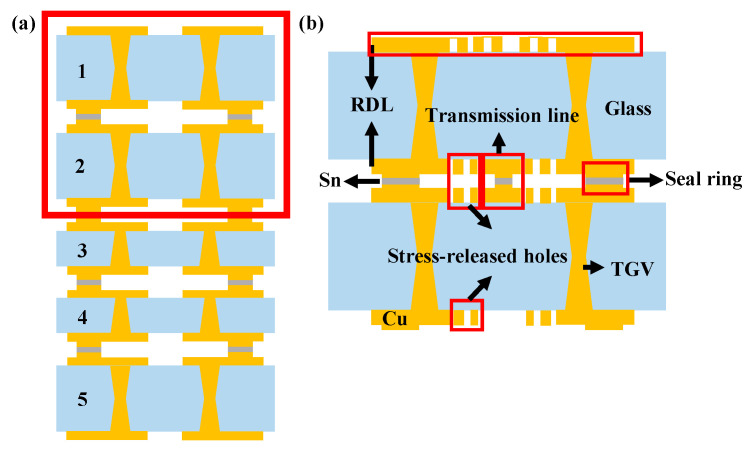
Glass lamination structure: (**a**) Five-layer lamination. (**b**) Double-layer lamination.

**Figure 10 micromachines-14-01149-f010:**
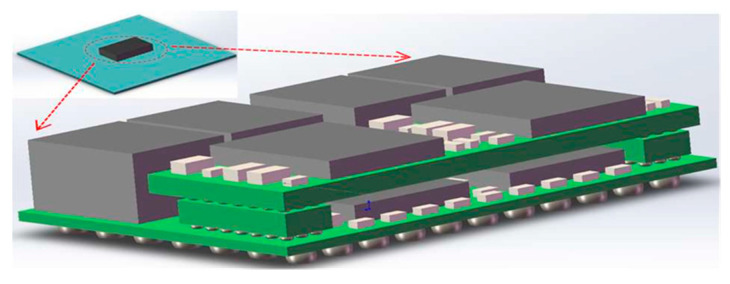
SiP structure schematic of power module.

**Figure 11 micromachines-14-01149-f011:**
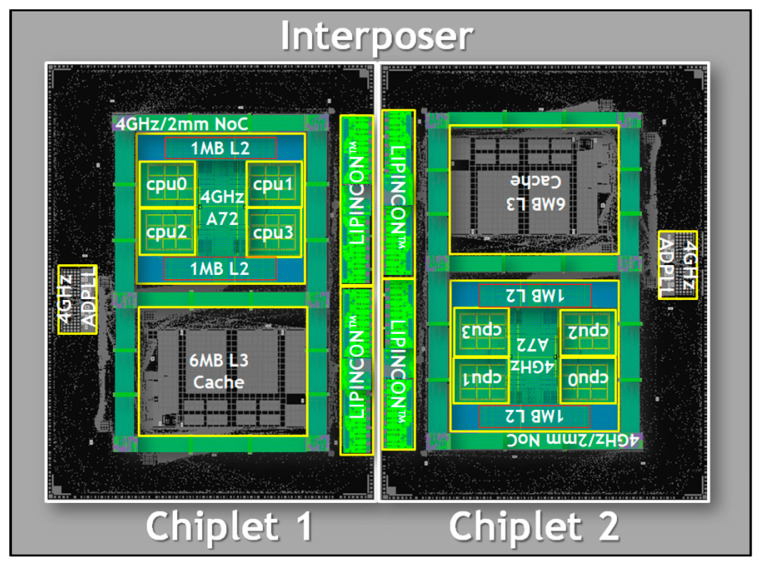
Schematic of the processor’s CoWoS package.

**Figure 12 micromachines-14-01149-f012:**
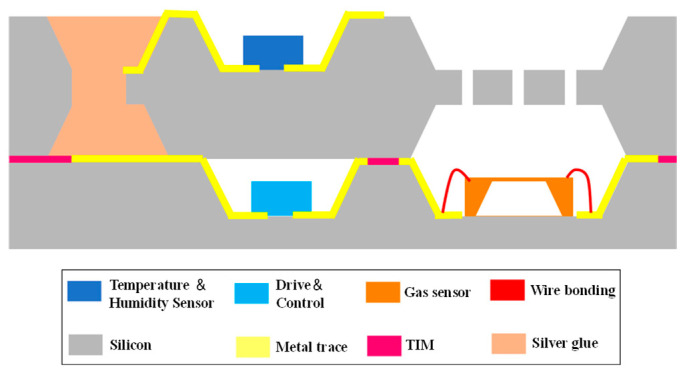
Cross-sectional diagram of the gas sensor package.

**Figure 13 micromachines-14-01149-f013:**
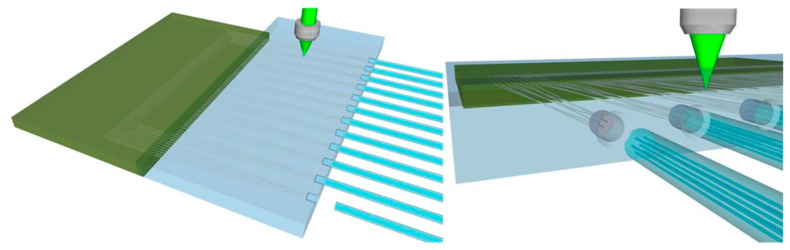
Femtosecond-laser-integrated package form.

**Figure 14 micromachines-14-01149-f014:**
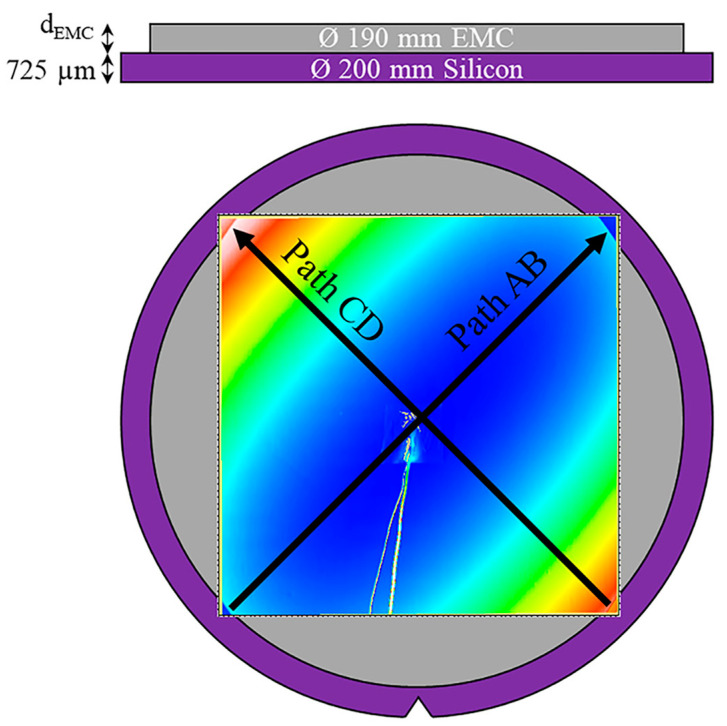
EMC warp demonstrator.

**Figure 15 micromachines-14-01149-f015:**
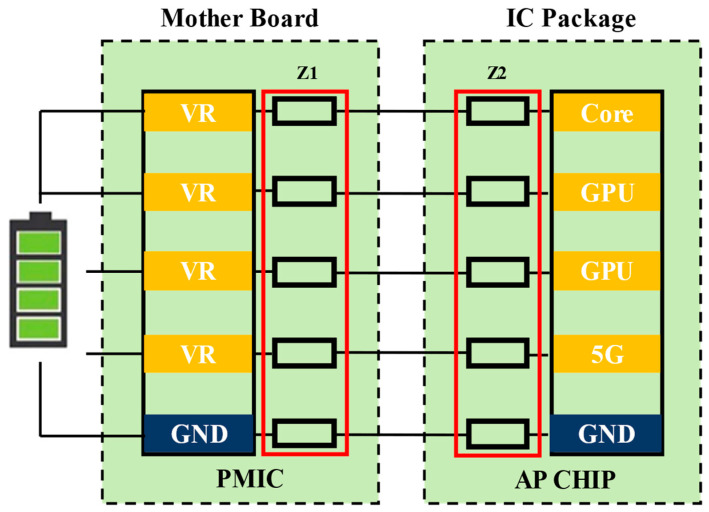
DSCP model based on RLC circuit.

**Figure 16 micromachines-14-01149-f016:**
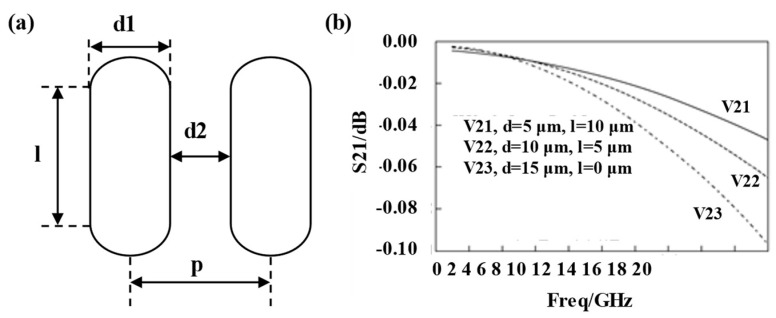
Research of strip TSV structure: (**a**) Schematic diagram of strip TSV structure. (**b**) Comparison of insertion loss of strip TSV.

**Figure 17 micromachines-14-01149-f017:**
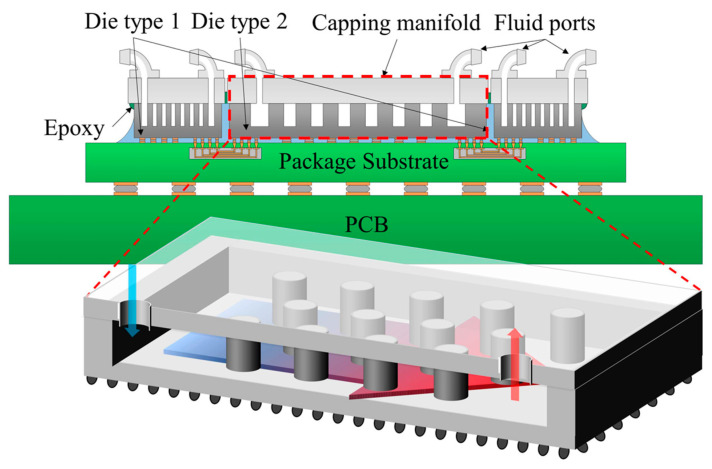
Schematic diagram of monolithic microfluidic cooling.

**Figure 18 micromachines-14-01149-f018:**
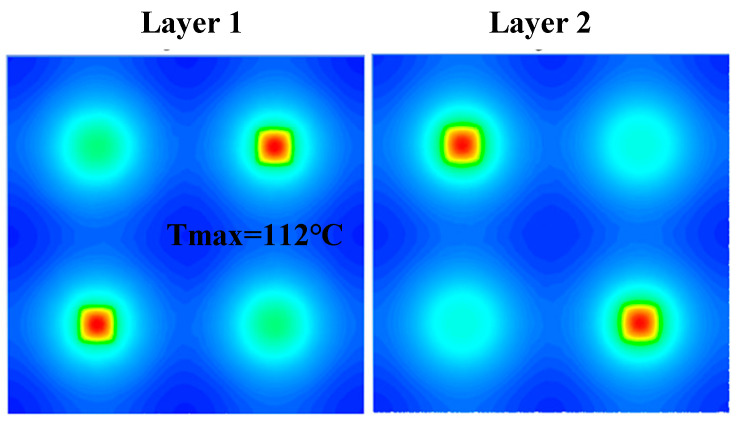
Temperature distribution diagram of optimized layout hotspots.

**Figure 19 micromachines-14-01149-f019:**
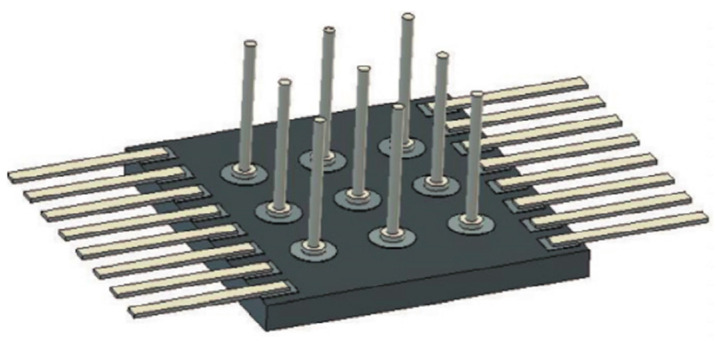
Tensile test structure.

**Figure 20 micromachines-14-01149-f020:**
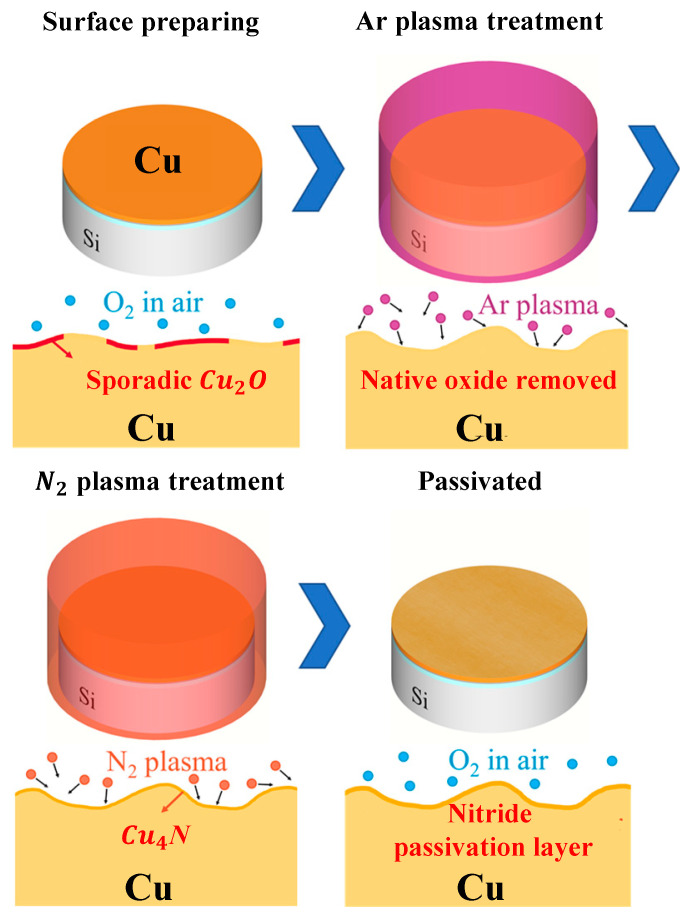
Two-step plasma treatment process.

**Figure 21 micromachines-14-01149-f021:**
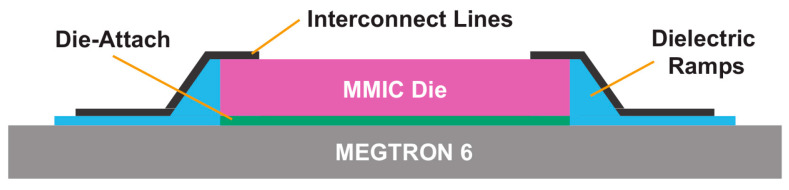
Schematic diagram of the two-dimensional cross-section of the ramp interconnection.

**Figure 22 micromachines-14-01149-f022:**
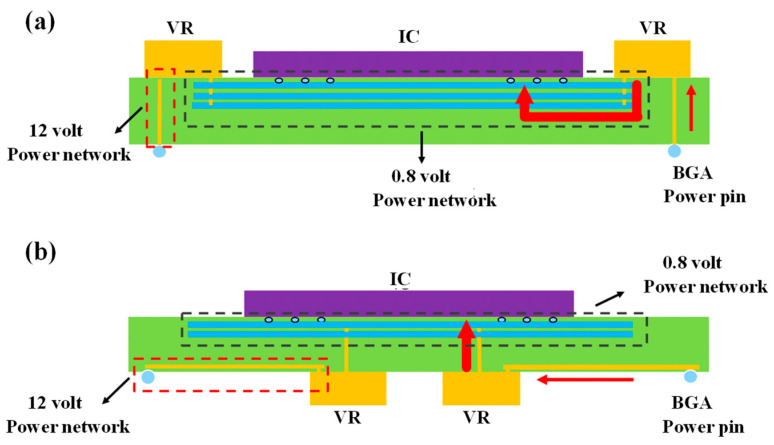
Topology of VR in SIP: (**a**) Top-level placement. (**b**) Bottom-level placement.

**Table 1 micromachines-14-01149-t001:** Comparison of the three levels in electronic integration.

Characteristic	SoC	SiP	PCB
Main material	Semiconductor	Conductor insulation	Conductor insulation
Volume	Small	Medium	Large
Cost	High	Low	Low
Period	Long	Short	Short
Type	One chip	Multiple chips	Multiple chips
Function	Simplification	Diversification	Diversification

**Table 2 micromachines-14-01149-t002:** Comparison of SiP forms.

Forms	Structure	Applications	Characteristics
FOWLP	2D	5G, AI [22,23,24]	Low cost Low parasitic effects High scalability
eWLB	2D/2.5D/3D	5G, AI [25,26,27,28]	Embedded Easy wiring Thin profile
INFO	2D	HPC, smartphones [24,29]	Multi-chip integration Systematization Multifunctional
DSMBGA	3D	RF, 5G [31]	Unique package structure Low power consumption Signal integrity
EMIB	2.5D	Graphics, HPC [32,33]	Good physical properties Easy processing Highly integrated

## Data Availability

No new data were created.

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
