# Peer review of "A Review of System-in-Package Technologies: Application and Reliability of Advanced Packaging"

_micromachines, 2023, doi:10.3390/mi14061149_

Round 1
Reviewer 1 Report
This is an excellent and thorough review on current situation and trend in the near future. However several figures needs to be refined, such as Fig.1, which may come from some kinds of EDA tools. (The figure 1, 3 and 9, which may be generated by some kind of EDA tools, are not clear enough, and therefore suggested to be refined.)
1. The submission is a thorough and well organized overview of current SiP research status.
2. The topic thus covered is relevant to the special issue and is of high value to the professionals engaged in the electronic packaging industry and device suppliers as well as R&D facilities involved.
3. The topic may be highly interesting to the readers of this journal considering the topics. However, the originality is hard to say. The perspective and organization of the submission are unique when compared to other overviews I have read.
4. The paper is well-written and the text is clear and easy to read.
5. The conclusions are consistent with the evidence and arguments presented and they address the main question posed.
Author Response
请参阅附件。

Reviewer 2 Report
please find it in the attachment.

Round 2
Reviewer 2 Report
please see the attachment.
